# Effects of Task Complexity on Linguistic Complexity for Sustainable EFL Writing Skills Development

**Liping Wang** [1] and **Chunlan Jin** [2,*]

1 The School of Foreign Languages, Nanchang University, Nanchang 330031, China; wangliping@ncu.edu.cn
2 The School of Foreign Languages, East China University of Science and Technology, Shanghai 200237, China
* Correspondence: jennyjin@ecust.edu.cn

**Abstract:** This paper reports the findings from a study that explored the effects of task complexity on linguistic complexity in EFL writing with a within-and-between-subject design for the sustainability of EFL writing skills. A total of 178 English majors and non-English majors participated in the study. They each performed two writing tasks that were manipulated using two variables: the number of elements and prior knowledge. Linguistic complexity was measured from both syntactic and lexical aspects with 39 indices. Data analysis indicated that task complexity produced different effects on different dimensions of linguistic complexity, suggesting that research on task complexity needs to break away from simplicity and that the Cognition Hypothesis and Limited Attentional Capacity Hypothesis require further verification and refinement. This study contributes to a better understanding of task complexity and linguistic complexity in EFL writing, yielding meaningful implications for pedagogy and assessment in this field.

**Keywords:** task complexity; number of elements; prior knowledge; linguistic complexity; sustainable development of writing skills

## 1. Introduction

Over the past decades, research on task complexity has featured primarily in the EFL writing papers, contributing to the sustainability of EFL writing skills. Most of the task complexity studies were conducted with guidance from the "Limited Attentional Capacity Hypothesis" and the "Cognition Hypothesis". Both of these hypotheses hold that attention resources play an important role in the task completion process, but their understandings of attentional resources are different, so their explanations are also different.

The "Limited Attentional Capacity Hypothesis" [1–3] proposes that learners' attentional capacity is limited, and the increase in task complexity will affect the complexity, accuracy, and fluency of the output language, resulting in a trade-off effect. The "Cognitive Hypothesis" [4–6], however, groups the factors that affect task complexity into two dimensions: resource-directing and resource-dispersing, and points out that these two dimensions exert different influences on the allocation of attentional resources: (1) increasing task complexity along the resource-directing dimension can direct attentional resources to specific language structures and forms, making the output language more accurate and complex, although its fluency will be reduced; (2) increasing task complexity along the resource-dispersing dimension will consume more attentional resources of learners and reduce their attention to language form, thereby affecting the accuracy, complexity, and fluency of the language; (3) the two dimensions of resource-directing and resource-dispersing will interact.

Therefore, the "Limited Attentional Capacity Hypothesis" and "Cognition Hypothesis" are not completely "tat-to-tat" [7], and their main difference lies in the resource-directing dimension. Although the research on task complexity is increasingly prevalent, the research focusing on both the resource-directing dimension and the resource-dispersing dimension

is still quite limited [8]. Therefore, this research simultaneously examines the two dimensions. Specifically, it explores the influences of two variables, the number of elements and prior knowledge, among which the number of elements belongs to the resource-directing dimension, and prior knowledge the resource-dispersing dimension. In fact, both of these two variables have been studied relatively extensively on their own according to Jonson's synthesis and meta-analysis of studies on cognitive writing task complexity [9], laying a solid basis for the research design of the present study. However, they have rarely been investigated together [9,10]. The present study assumes that role. The simultaneous study of the number of elements and prior knowledge will not only further verify the Limited Attentional Capacity Hypothesis and the Cognition Hypothesis by analyzing the main effects of the two variables respectively, but also contribute to the improvement of the Cognition Hypothesis by demonstrating how resource-directing and resource-dispersing factors interact with each other. Moreover, it will also provide insights into L2 learners' processing capacity, as well as task-based teaching and assessment.

## 2. Literature Review

### 2.1. Studies on the Number of Elements of Writing Tasks

　　Kuiken and his colleagues conducted pioneering studies of writing task complexity manipulated by a number of elements. In their study [11], the elements used were the requirements taken into account when choosing a holiday destination: six in the complex task and three in the non-complex task. Linguistic complexity was measured from both syntactic and lexical aspects with indices such as the number of clauses per T-unit (hereafter C/T), the number of dependent clauses per clause (hereafter DC/C), and the number of word types divided by the total number of word tokens (hereafter WT/W). A total of 62 junior students of Italian at the University of Amsterdam completed both the more-element and the fewer-element tasks in 1 session. Data analysis indicated that there was "no significant difference in task complexity on the syntactic complexity or lexical variation in linguistic performance" [11] (p. 210).

　　Similar to Kuiken and his colleagues, Frear and Bitchener [12] designed a restaurant-choosing task. In the medium-complexity writing task, participants needed to choose one of the two restaurants for their friend and justify their choice based on the information about each restaurant, as well as about their friend. In the high-complexity writing task, the number of candidate restaurants and friends both amounted to three, increasing the elements in the task. However, in contrast to Kuiken and his colleagues, Frear and Bitchener [12] designed patently a low-complexity task by asking the participants to write to their friend who was coming to New Zealand and tell them about New Zealand. A total of 34 non-native intermediate-level speakers of English participated in the study, and they completed all 3 tasks in 2 stages within 1 week. The linguistic complexity measures used in this study were a mean segmental type-token ratio (hereafter MSTTR) and the number of dependent clauses per T-unit (hereafter DC/T), which were analyzed across all dependent clauses, including adjective dependent clauses, nominal dependent clauses, and adverbial dependent clauses. Findings revealed that increasing task complexity, with respect to the number of elements, produced a significantly greater number of adverbial dependent clauses per T-unit and more lexical variation.

　　The results from the above study were not consistent with those from Cho's study, although they were both published in 2015. Cho [13] recruited 110 Korean high school students with various English proficiency to complete writing tasks of different complexities manipulated by the number of elements, represented by the number of roommate candidates. Each participant completed only one task, either the simple or the complex. Linguistic complexity was only measured syntactically, using C/T and the number of T-units per sentence (hereafter T/S). Results showed that the number of elements involved in the writing task had no significant effect on syntactic complexity.

　　Rahimi and Zhang [14] also examined the effects of the number of elements on linguistic complexity by designing 2 tasks of 90 min total duration. For their cognitively

simple task, they asked 80 upper-intermediate L2 English learners to write about how to allocate $5,000,000 to 3 competing projects. For the cognitively complex task that followed, $10,000,000 had to be allocated across 6 competing projects in 90 min. Five linguistic complexity measures were used: the mean length of clauses (hereafter MLC), DC/C, co-ordinate phrases per T-unit (hereafter CP/T), D value, and lexical sophistication. Data analysis indicated that the more elements a writing task had, the more dependent clauses and sophisticated words were used. However, MLC, CP/T, and the D value were not significantly influenced by the number of elements involved in the writing task.

Lee [15] designed a best location task in his study of task complexity in which elements were increased in terms of three numbers: the number of the locations to choose from, the number of pieces of information about the locations, and the number of requirements involved in the task. Written performance was measured from syntactic and lexical aspects, using indices such as the mean length of T-units (hereafter MLT), subordinate clauses per T-unit (SC/T), and the G value. Results showed that increasing task complexity along the number of elements produced no significant influence on syntactic complexity but led to greater lexical variation.

### 2.2. Studies on Prior Knowledge of Writing Tasks

Prior knowledge, a term originating in psychology, is closely related to Schema Theory [16]. It is, to a large degree, equal to topic familiarity in earlier studies [17]. To judge the participants' prior knowledge of a writing task is to ascertain whether they are familiar with the topic. The more familiar they are with the task topic, the less complex the task is.

Adams and Nik [18] examined the effects of prior knowledge on second language production in a text-based chat by employing a problem-solving task. Two groups of students participated in the study. One group was majoring in electrical engineering, and the other in chemical engineering. They were required to role-play engineers in a multinational company meeting online to decide which type of electrical engineering software the company should adopt. Although each participant was given enough details about their assigned software, the students majoring in electrical engineering were supposed to be more familiar with the task. Four indices were used to measure the syntactic and lexical complexity of the participants' output, namely the mean number of clauses per AS-unit (C/AS), the mean number of words per turn (W/T), the Lexical Frequency Profile (LFP), and the Guiraud's Index. Data analysis indicated that students produced significantly higher lexical complexity in the task with less prior knowledge.

Yang [19] designed a writing task in which participants discussed the benefits and possible problems that computers and the Internet brought to the writer as a university student, to university students in the writer's country, and to people in underdeveloped areas of the world, respectively. Prior knowledge of the writing topic was controlled by the target people that computers and the Internet have influenced. Eleven linguistic complexity measures were employed, including the mean length of a sentence (hereafter MLS), MLT, MLC, T/S, DC/T, non-finite elements per clause, coordinate phrases per verb phrase, complex noun phrases per verb phrase, LS1, vocd D, as well as the number of lexical words out of the total words. Data analysis showed that prior knowledge of the writing topic had almost no significant effect on syntactic complexity, but it did have a significant effect on lexical complexity. "The lexical diversity and lexical sophistication of the essays on the lower familiarity task were significantly lower than those of the essays on the other two more familiar tasks" [19] (p. 118).

In conclusion, there are inconsistencies in the results of the existing studies investigating the number of elements and prior knowledge of writing tasks. For example, as to the effect of the number of elements on syntactic complexity, two studies [12,14] reported a significant influence, while the other three studies in this respect [11,13,15] found no significant effect. As to the effects of the number of elements on lexical complexity, only one study [11] reported no significant effect. In all of the other three studies [12,14,15], more elements in the writing task did produce greater lexical complexity, but in the different sub-

dimensions of lexical complexity, lexical variation [12,15], and lexical sophistication [14], respectively. Studies of prior knowledge are smaller in number, but they also presented contradictory results. Adams and Nik [18] found that increasing task complexity along with prior knowledge significantly promoted lexical complexity. This goes against Yang's result [19], which indicated that in the more complex task with less prior knowledge, students produced significantly lower lexical complexity.

The inconsistencies of the research results could be attributed to the following three reasons: (1) some of the writing tasks employed in some studies might not be of different cognitive complexity, such as the restaurant-choosing task, which is not necessarily more complex than introducing New Zealand [12]; (2) in most of the previous studies, only a limited number of indices were used to measure the linguistic complexity, which may present only some dimensions of the linguistic complexity, not the whole picture; (3) the indices used in previous studies are inconsistent (such as C/T and DC/C in Kuiken's study [11] compared to DC/T in Frear and Bitchener's [12]), which might lower the comparability of the research results. Therefore, this study seeks to improve the task design to ensure that the writing tasks are truly of various complexity. Moreover, it measures linguistic complexity with more indices (39 altogether) in order to improve the comparability of results among studies.

With the above considerations in mind, this study set out to address the following research questions with a within-and-between-subject design:

(1)   What are the main effects of the number of elements and prior knowledge on the linguistic complexity of the written text?
(2)   What are the interactive effects of the number of elements and prior knowledge on the linguistic complexity of the written text?

### 3. Methods

#### 3.1. Participants

This study employed a mixed design of within-and-between subjects. Each subject needed to complete two English writing tasks in different time periods. In other words, this study had higher requirements for the cooperation of the research subjects. In view of this, 178 undergraduates from 4 parallel intact classes of English majors ($N = 71$) and 4 parallel intact classes of non-English majors ($N = 107$) at the university where one of the researchers was working were recruited on the basis of convenience sampling, with an average age of 19.3 years.

Although the English majors came from four different classes, they were homogeneous to a large extent since they usually shared the same teacher for the same course. Prior to the start of the study, the researchers conducted a one-way analysis of variance on the paper scores of the four classes in their most recent comprehensive English test before the start of the study. Data analysis showed that there was no significant difference in the English proficiency of the four classes (see Table 1 for details).

**Table 1.** Comprehensive English Test Scores Comparison.

| Major | Grade | No. of Subjects | M ± SD | | | | One-Way Analysis of Variance | |
|---|---|---|---|---|---|---|---|---|
| | | | Group 1 | Group 2 | Group 3 | Group 4 | F | p |
| English | 2 | 71 | 74.29 ± 8.59 | 77.53 ± 7.80 | 71.71 ± 14.03 | 73.40 ± 10.39 | 0.934 | 0.429 |
| Non-English | 2 | 107 | 74.03 ± 5.68 | 73.51 ± 8.10 | 73.70 ± 5.78 | 75.17 ± 7.27 | 0.294 | 0.829 |

The non-English majors likewise came from four different classes, covering majors such as history, e-commerce, business administration, and electrical engineering. Before the study, the researchers performed a one-way analysis of variance on their most recent college English test and found that the English proficiency of the four classes was equivalent (see Table 1 for details). In addition, the four classes had the following elements in common: (1) they shared the same teacher for their College English course, who was a female lecturer

of 39 years of age majoring in foreign linguistics and applied linguistics, with 17 years of teaching experience in colleges and universities; (2) they were all students from English Class A, with relatively high English proficiency; (3) they were all preparing for CET-4 (the full name is College English Test 4, a national English proficiency test for non-English majors in China) when invited to perform the first writing task, and all believed that this was a good opportunity to practice their English writing.

The four classes of English majors and the four classes of non-English majors merged into four new groups of participants to perform the four writing tasks in this study, with one new group consisting of one class of English majors and one class of non-English majors. It was deemed insufficient to simply compare English proficiency among the four new groups by only providing separate scores for English and non-English majors instead of proficiency indices for the whole group. In fact, we, at first, did plan to organize an English writing test for both English and non-English majors to assess and compare their English writing proficiency as a whole group. However, this plan was later abandoned, considering that participants in this study would have to complete two writing tasks, which would have been a huge challenge for them. Another writing task would scare away some participants. Moreover, no significant differences between the four English majors' classes and the four non-English majors' classes would, to a large extent, guarantee the homogeneity of the four new groups of participants in terms of English proficiency, since every new group was comprised of both English and non-English majors.

*3.2. Instruments*

The instruments of this study consisted of four English writing tasks (see Supplementary Materials S1): task 1 was characterized by fewer elements and more prior knowledge; task 2 had fewer elements and less prior knowledge; task 3 had more elements and more prior knowledge; task 4 more elements and less prior knowledge. According to the task complexity framework [6], among the above four English writing tasks, task 1 was the simplest, task 4 was the most complex, and task 2 and task 3 were of medium complexity.

Specifically, this study designed a life partner-choosing task. The number of candidates and the number of features each candidate possessed were the "elements" of the task. In the task with fewer elements, two candidates were different in "talent" and "diligence", while in the task with more elements, there were three candidates who were different in terms of the following four characteristics: "talent", "diligence", "social ability", and "family background". As for "prior knowledge", the simple and complex tasks differentiated in whether participants were supposed to look for their own life partner and justify their preferences or were required to choose one life partner that their parents might recommend to them and justify their parents' preferences with detailed reasons.

*3.3. Data Collection and Analysis*

This study employed a two-factor mixed design, which is a type of repeated measurement. The number of elements and prior knowledge were randomly assigned to be the within-subject factor and the between-subject factor, respectively. In fact, the most ideal experimental design is a two-factor within-subject design, considering it "can diminish all the variations caused by individual differences in the test, hence reducing experimental errors and improving the accuracy of the results" [20] (p. 101). However, if a two-factor within-subject design was adopted, each subject would have to complete four English essays with just a few differences in the prompts, which might cause fatigue to the participants and lead to practice effects [20], as well as low feasibility. In view of this, this study employed a two-factor mixed design. Although the results of the between-subject factor in this design were not as accurate as those of the two-factor within-subject design, the accuracy of "the within-subject and the interactive effects of the two factors are both good" [20] (p. 93), and the control of individual differences in the test was superior to that of the two-factor between-subject design.

According to the experimental design of this study, the data was collected in May and September 2019. May and September were selected deliberately for two reasons: (1) the four-month interval between them would reduce the impact of task repetition [21,22] to a large extent; (2) the two-month summer vacation from July to August would help control other variables, such as instruction and practice, which might lead to improvements in writing performance, because the majority of participants were on vacation. In addition, this study also adopted the "ABBA method" (see Table 2 for details) to balance the systematic errors caused by the presentation order of writing tasks [23]. A and B, respectively, represented two levels of experimental treatment, which referred to the "fewer" and "more" elements in this study. For example, English major Class 1 completed the tasks with fewer elements (A) and more elements (B) in the first and second phases of data collection, respectively, while English major Class 2 completed the tasks in the reverse order.

**Table 2.** Data Collection Arrangement.

| Data Collection Phase | Data Collection Time | Subjects | | | Writing Tasks | | Experimental Operation Level |
|---|---|---|---|---|---|---|---|
| | | Major | Class | Number | Type | Task Complexity | |
| 1 | May 2019 | English | 1 | 17 | Task 1 | fewer E *, more P * | A |
| | | | 2 | 17 | Task 3 | more E *, more P * | B |
| | | | 3 | 17 | Task 2 | fewer E *, less P * | A |
| | | | 4 | 20 | Task 4 | more E *, less P * | B |
| | | Non-English | 5 | 29 | Task 1 | fewer E *, more P * | A |
| | | | 6 | 28 | Task 3 | more E *, more P * | B |
| | | | 7 | 27 | Task 2 | fewer E *, less P * | A |
| | | | 8 | 23 | Task 4 | more E *, less P * | B |
| 2 | September 2019 | English | 1 | 17 | Task 3 | more E *, more P * | B |
| | | | 2 | 17 | Task 1 | fewer E *, more P * | A |
| | | | 3 | 17 | Task 4 | more E *, less P * | B |
| | | | 4 | 20 | Task 2 | fewer E *, less P * | A |
| | | Non-English | 5 | 29 | Task 3 | more E *, more P * | B |
| | | | 6 | 28 | Task 1 | fewer E *, more P * | A |
| | | | 7 | 27 | Task 4 | more E *, less P * | B |
| | | | 8 | 23 | Task 2 | fewer E *, less P * | A |

* E = elements; P = prior knowledge.

The time limit for each writing task was the same: 40 min. The same group of participants completed the same writing task. During the writing process, they could not communicate with each other or use mobile phones, dictionaries, and other tools. English and non-English majors were required to write at least 250 and 150 words, respectively, which is consistent with the popular practices in their daily English learning. After 40 min, the task stopped, and all the written texts were collected.

All the collected texts were then converted into MS word documents without any error correction, except for the spelling ones, which were changed following Yu's practice in his study in 2009: (1) if a misspelt word was correct elsewhere, it was corrected, otherwise, it was entered as it was; (2) if the same word was spelt incorrectly and differently in different places (≥2), one of the wrong spellings was randomly chosen to replace all the others. The reason for the changes to the misspelt words was that the inclusion of two or more misspelt forms of the same word might have increased the lexical diversity of the text [24].

After all the word documents were prepared, they were processed with the automatic syntactic and lexical complexity analyzers to measure their syntactic and lexical complexity. The syntactic complexity analyzer has 14 indices that may be categorized into five types, namely, length of the production unit, sentence complexity, subordination, coordination, and particular structures [25]. The first type, the length of the production unit, consists of three indices that measure the length at the clausal, sentential, and T-unit levels (MLC, MLS, MLT). The second type, sentence complexity, is gauged by the number of clauses per sentence (C/S). The third type, subordination, consists of four ratios, namely, the T-unit complexity ratio (C/T), the complex T-unit ratio (CT/T), and two dependent ratios (DC/C, DC/T). The fourth type, coordination, is measured with the ratio of coordinate clauses

(CP/C, CP/T) and the ratio of T-units per sentence (T/S). The last type, particular structures, examines the ratios of particular syntactic structures such as complex nominals and verb phrases against larger production units (CN/C, CN/T, VP/T). Among these 14 indices, 11 have proved to be "significantly correlated to language proficiency, second language development or writing quality" [26]. The remaining three indices were recommended for further investigation by Wolfe-Quintero [27]. The syntactic complexity analyzer has widely been used in studies of second language writing [28–32].

The lexical complexity analyzer includes 25 measures covering lexical density, lexical sophistication, and lexical variation [33]. Lexical density (LD) is measured by the ratio of lexical words to total words in a text. Lexical sophistication measures the "the proportion of relatively unusual or advanced words in the learner's text" [34] (p. 203), with five indices from prior studies, namely LS1 [35], LS2 [36], VS1 [37], CVS1 [27], and VS2 [38]. Lexical variation, also known as lexical diversity, refers to "the range of a learner's vocabulary as displayed in his or her language use" [33] (p. 192), measured by 19 indices. Among them, four indices gauge the number of different words in a language sample (NDW, NDW-50, NDW-ER50, NDW-ES50), five relate to the type-token ratio (TTR, MSSTTR-50, CTTR, RTTR LogTTR), nine measure the variation of different kinds of words (LV, VV1, SVV1, CVV1, VV2, NV, AdjV, AdvV, ModV), and one is an uber index. The Stanford POS Tagger and Morpha Lemmatizer used by this lexical complexity analyzer are of 95% accuracy and 99% accuracy, respectively, ensuring the accuracy of lexical complexity analysis (personal correspondence). The lexical complexity analyzer is also gaining popularity in second language writing research [19,32,39].

## 4. Results and Discussion

In this section, the research results will be reported corresponding to the two research questions of this study. We will first present the main effects of the number of elements and prior knowledge on linguistic complexity, respectively, before discussing the interactive effects of these two variables on linguistic complexity.

### 4.1. The Main Effects of the Number of Elements on Linguistic Complexity

Table 3 summarizes the descriptive statistics for different linguistic complexity measures across four English writing tasks. A series of two-way repeated-measures ANOVA was conducted after the data were checked for the assumptions of normality and sphericity. Results showed that the number of elements exerted significant effects on ten linguistic complexity measures (see Table 4).

**Table 3.** Descriptive statistics for linguistic complexity measures across writing tasks (The table only lists measures with *p* values less than 0.05 in the two-factor repeated measures ANOVA. For data on all the measures, please see Supplementary Materials S2).

| Linguistic Complexity | Measures | Task 1 | | Task 2 | | Task 3 | | Task 4 | |
|---|---|---|---|---|---|---|---|---|---|
| | | M | SD | M | SD | M | SD | M | SD |
| Syntactic complexity | MLS | 15.196 | 3.125 | 15.508 | 3.861 | 16.045 | 3.704 | 16.731 | 4.036 |
| | T/S | 1.104 | 0.119 | 1.129 | 0.140 | 1.149 | 0.175 | 1.150 | 0.176 |
| | CN/T | 1.407 | 0.418 | 1.458 | 0.441 | 1.301 | 0.449 | 1.281 | 0.421 |
| | C/S | 2.077 | 0.410 | 2.116 | 0.475 | 2.186 | 0.513 | 2.204 | 0.514 |
| Lexical comlexity | LD | 0.499 | 0.032 | 0.492 | 0.031 | 0.502 | 0.031 | 0.503 | 0.026 |
| | LS1 | 0.284 | 0.056 | 0.259 | 0.063 | 0.262 | 0.055 | 0.248 | 0.053 |
| | NDW | 112.677 | 24.454 | 106.388 | 24.520 | 116.860 | 26.556 | 111.318 | 23.399 |
| | NDW-50 | 33.409 | 3.398 | 35.047 | 3.327 | 35.452 | 3.484 | 35.529 | 3.594 |
| | TTR | 0.457 | 0.057 | 0.469 | 0.056 | 0.463 | 0.057 | 0.454 | 0.060 |
| | MSTTR-50 | 0.728 | 0.041 | 0.732 | 0.044 | 0.745 | 0.041 | 0.740 | 0.039 |
| | VV1 | 14.989 | 5.704 | 13.465 | 4.734 | 14.122 | 5.143 | 12.829 | 4.744 |
| | SVV1 | 2.688 | 0.523 | 2.552 | 0.468 | 2.614 | 0.484 | 2.491 | 0.465 |
| | VV2 | 0.160 | 0.033 | 0.164 | 0.035 | 0.154 | 0.031 | 0.144 | 0.030 |
| | NV | 0.556 | 0.088 | 0.561 | 0.093 | 0.570 | 0.098 | 0.530 | 0.088 |

**Table 4.** The main and interactive effects of the number of elements and prior knowledge on linguistic complexity (The table only lists measures with *p* values less than 0.05 in the two-factor repeated measures ANOVA. For data on all the measures, please see Supplementary Material S2).

| Linguistic Complexity | Measures | Number of Elements | | | Prior Knowledge | | | Number of Elements Prior Knowledge | | |
|---|---|---|---|---|---|---|---|---|---|---|
| | | *F* | *df* | *p* | *F* | *df* | *p* | *F* | *df* | *p* |
| Syntactic complexity | MLS | 8.518 | 1 | 0.004 | | | | | | |
| | T/S | 4.399 | 1 | 0.037 | | | | | | |
| | CN/T | 10.186 | 1 | 0.002 | | | | | | |
| | C/S | 4.739 | 1 | 0.031 | | | | | | |
| Lexical complexity | LD | 5.617 | 1 | 0.019 | | | | | | |
| | LS1 | 7.724 | 1 | 0.006 | 10.754 | 1 | 0.001 | | | |
| | NDW | 6.074 | 1 | 0.015 | | | | | | |
| | NDW-50 | 13.461 | 1 | 0.000 | 4.913 | 1 | 0.028 | 5.141 | 1 | 0.025 |
| | TTR | | | | | | | 5.050 | 1 | 0.026 |
| | MSTTR-50 | 9.436 | 1 | 0.002 | | | | | | |
| | VV1 | | | | 4.817 | 1 | 0.029 | | | |
| | SVV1 | | | | 4.487 | 1 | 0.036 | | | |
| | VV2 | 19.440 | 1 | 0.000 | | | | 5.478 | 1 | 0.020 |
| | NV | | | | | | | 7.691 | 1 | 0.006 |

Specifically, in terms of syntactic complexity, learners used significantly longer sentences when completing the more complex task with more elements (MLS, $F_{(1, 176)} = 8.518$, $p = 0.004$). In addition, the number of T units (T/S, $F_{(1, 176)} = 4.399$, $p = 0.037$) and the number of clauses (C/S, $F_{(1, 176)} = 4.739$, $p = 0.031$) in each sentence in the more complex task with more elements were also higher than those in the less complex task with fewer elements. However, compared with the less complex task with fewer elements, the number of complex nominals per T unit (CN/T) in the more complex tasks with more elements was significantly smaller ($F_{(1, 176)} = 10.186$, $p = 0.002$). In other words, increasing writing task complexity based on the number of elements significantly increased the length of the production unit, the number of coordinates, and the sentence complexity of the written texts, while it significantly reduced the number of particular structures used in the written text.

In terms of lexical complexity, the lexical density (LD, $F_{(1, 176)} = 5.617$, $p = 0.019$), the number of different words (NDW, $F_{(1, 176)} = 6.074$, $p = 0.015$), the number of different words in the first 50 words of the written texts (NDW-50, $F_{(1, 176)} = 13.461$, $p = 0.000$), and the mean segmental type-token ratio of the first 50 words in the written texts (MSTTR-50, $F_{(1, 176)} = 9.436$, $p = 0.002$) produced in the more complex task with more elements were all significantly higher than those in the less complex task with fewer elements. However, the number of sophisticated lexical words (LS1, $F_{(1, 176)} = 7.724$, $p = 0.006$) and verb variation (VV2, $F_{(1, 176)} = 19.440$, $p = 0.000$) in the more complex task with more elements were significantly smaller than those in less complex task with fewer elements. In other words, increasing writing task complexity based on the number of elements significantly increased the lexical density, the number of different words, and the type-token ratio of the written texts, while it significantly reduced the lexical sophistication and verb variation.

The results that increasing task complexity, with respect to the number of elements, significantly produced longer production units and greater coordination ratios support the Cognition Hypothesis but are not consistent with the findings of Rahimi and Zhang's study [14], which found that task complexity produced no significant effect on production units or coordination. This might be attributed to participants' different English proficiency in these two studies. Participants recruited by Rahimi and Zhang [14] were upper-intermediate L2 English learners, while in this study, participants only had limited English knowledge since they were made up of first-year non-English majors and second-year English majors. Writers of different proficiency might use different syntactic structures. Specifically, writers of low proficiency tend to use a lot of coordination [40], while those of high proficiency favor nominalization and participle modifiers [41], which would shorten

the sentences. Moreover, the more frequent use of coordinate structures might contribute to the greater sentence complexity in this study which was measured by calculating the number of clauses per sentence.

Increasing task complexity, with respect to the number of elements, significantly reduced the number of particular structures, that is, complex nominals, used in the written texts. This result is not consistent with the Cognition Hypothesis but supports the Limited Attentional Capacity Hypothesis and is of great significance since most relevant studies in this respect [11–15] did not examine nominals or phrases. However, phrasal complexity could act together with production units, subordination, and coordination to present the whole picture of syntactic complexity [42]. The development from coordinate structures to subordinate structures and then to compacted sentences represents grammatical complexification [41]. In compacted sentences, there are fewer relative clauses but more adjective and adverbial phrases [43], as well as more infinitives and nominals [41]. Therefore, syntactic complexity ought to be measured from not only the length of the production units and the number of coordinate and subordinate clauses but also from the complexity of phrases. More studies on particular structures are expected.

Increasing writing task complexity, with respect to the number of elements, significantly increased lexical density. This result supports the Cognition Hypothesis and enriches current findings about lexical complexity influenced by task complexity manipulated by the number of elements because most of the previous studies [11–15] did not examine lexical density, as is the case with particular structures discussed in the preceding paragraph. Lexical density, in this study, refers to "the ratio of the number of lexical words to the total number of words in a text" [33] (p. 191). The greater lexical density a sentence has, the more lexical words are used in the sentence and, resultingly, more information is delivered since sentence information is largely expressed by its lexical words [44]. This study found that learners used more lexical words when they performed the more complex writing tasks with more elements. This might be because learners had more information to exchange, prompted by more elements in the task.

Increasing writing task complexity, with respect to the number of elements, significantly increased the type-token ratio. These results support the Cognition Hypothesis, Frear and Bitchener [12], and Lee [15], but not Kuiken et al. [11] or Rahimi and Zhang [14]. The reasons for the inconsistency in the research results might lie in the different indices used in these studies. Specifically, the present study used six indices to measure the type-token ratio, namely TTR, MSTTR-50, CTTR, RTTR, LogTTR, and Uber. However, in the other studies, only one or two indices were used: for example, TTR and CTTR in Kuiken et al. [11], D in Rahimi and Zhang [14], and G in Lee [15]. The present study also used TTR and CTTR, which were not influenced by task complexity. Therefore, in this sense, the present study also verifies Kuiken's research results. Moreover, the previous studies did not report on the number of words, which was indicated in this study to increase alongside the number of elements in the writing task. This is understandable considering that the type-token ratio also increased.

Increasing writing task complexity, with respect to the number of elements, significantly reduced the lexical sophistication. In previous relative studies, only Rahimi and Zhang [14] measured lexical sophistication. However, their results, which indicated that learners used fewer sophisticated words in writing tasks with fewer elements, are not consistent with ours. This might be due to the fact that these two studies have different operational definitions for sophisticated words. In the present study, a word was sophisticated if it was "not on the list of the 2000 most frequent words generated from the British National Corpus" [33] (p. 192). Rahimi and Zhang [14], however, used Range 32 and defined a word as sophisticated if it was on the third list. Therefore, future studies should not only employ the same measures, but also the same operational definition for the same measures. Only in this way can the research results be compared.

Increasing writing task complexity, with respect to the number of elements, significantly reduced verb variation, which was not studied in previous relative studies [11–15].

A greater number of elements involved in the writing task resulted in fewer diversified verbs being used. It is apparent that task complexity and verb variation were competing for the writers' limited attentional capacity, which is consistent with the Limited Attentional Capacity Hypothesis.

### 4.2. The Main Effects of Prior Knowledge on Linguistic Complexity

Tables 3 and 4 also reveal that prior knowledge had significant effects on four linguistic complexity measures. Compared with the more complex task with less prior knowledge, the number of different words in the first 50 words (NDW-50) of the written texts in the less complex task with more prior knowledge was significantly smaller ($F = 4.913$, $p = 0.028$), while its lexical sophistication (LS1) and verb variation (VV1, SVV1) were both significantly higher ($F_{LS1} = 10.754$, $p_{LS1} = 0.001$; $F_{VV1} = 4.817$, $p_{VV1} = 0.029$; $F_{SVV1} = 4.487$, $p_{SVV1} = 0.036$). In other words, increasing writing task complexity based on the prior knowledge variable significantly increased the number of different words in the written text but, at the same time, significantly reduced the lexical sophistication and verb variation of the written text.

Increasing writing task complexity, with respect to prior knowledge, significantly increased the number of different words. This result challenges the Limited Attentional Capacity Hypothesis, as well as the Cognition Hypothesis, which held that lexical complexity would be reduced when the task complexity was increased. In fact, the simplest way to measure the number of different words in a text is to calculate its type-token ratio, but the type-token ratio is easily influenced by the length of the text [45]. In view of this, the present study added three measures, namely, NDW-50, NDWER-50, and NDWES-50. Data analysis showed that when writers had less prior knowledge of the writing topic, they produced a greater number of different words in the first 50 words of their written text. This result could be supplementary to the findings on prior knowledge of the writing task, considering that none of the existing studies in this respect [18,19] measured the number of different words.

Increasing writing task complexity, with respect to prior knowledge, significantly reduced the lexical sophistication of the written text. This result is consistent with Yang's [19] findings. When writers had less prior knowledge of the writing task, they would pay more attention to the task and less attention to the language, which might result in their use of more frequently used words in their writing. The same reason may also explain why verb variation was smaller when learners performed the more complex writing task with less prior knowledge. If learners had paid more attention to their search for different verbs to express their ideas, they might not have completed the writing task in time. These results concerning lexical sophistication and verb variation support the Limited Attentional Capacity Hypothesis and the Cognition Hypothesis.

### 4.3. The Interactive Effects of the Number of Elements and Prior Knowledge on Linguistic Complexity

As is shown in Tables 3 and 4, the number of elements and prior knowledge had significant interactive effects on 4 linguistic complexity measures of the written texts, including the number of different words in the first 50 words (NDW-50, $F (1, 176) = 5.141$, $p = 0.025$), the type-token ratio (TTR, $F (1, 176) = 5.050$, $p = 0.026$), verb variation (VV2, $F (1, 176) = 5.478$, $p = 0.020$), and noun variation (NV, $F (1, 176) = 7.691$, $p = 0.006$). However, Tables 3 and 4 cannot show the effects of the number of elements on different levels of prior knowledge, and vice versa. Therefore, a simple effect analysis was conducted.

The simple effect analysis results indicate (see Table 5):

(1)  When the learner had more relevant prior knowledge, the number of elements involved in the task exerted no significant effect on the type-token ratio (TTR), verb variation (VV2), and noun variation (NV) of the written texts. However, when the learner had less relevant prior knowledge and there were fewer elements involved in the task, the type-token ratio, verb variation, and noun variation of the written texts were all higher;

(2) When the learner had less relevant prior knowledge, the number of elements involved in the task had no significant effect on the number of different words in the first 50 words of the written text (NDW-50), but when the learner had more relevant prior knowledge and there were fewer elements involved in the task, the number of different words in the first 50 words of the written text was smaller.

**Table 5.** Results of simple effect analysis.

| Vocabulary Complexity | Coding | MD & *p* Value | More Prior Knowledge | | Less Prior Knowledge | |
|---|---|---|---|---|---|---|
| | | | Fewer Elements | More Elements | Fewer Elements | More Elements |
| Vocabulary diversity | NDW-50 | MD | −1.943 | | −0.482 | |
| | | *p* | 0.000 | | 0.034 | |
| | TTR | MD | −0.005 | | 0.015 | |
| | | *p* | 0.385 | | 0.024 | |
| | VV2 | MD | 0.006 | | 0.020 | |
| | | *p* | 0.136 | | 0.000 | |
| | NV | MD | −0.014 | | 0.031 | |
| | | *p* | 0.210 | | 0.009 | |

Note: MD is short for "Mean Difference".

In addition, it can be deduced from Tables 3 and 4 that the number of elements and prior knowledge had no significant interactive effects on unit length, subordination, coordination, particular structures, lexical density, or lexical sophistication.

The above results could be interpreted as when writers were not familiar with the writing topic, increasing task complexity with more elements reduced the lexical variation of the written text, but when writers were familiar with the writing topic, increasing task complexity with more elements promoted the lexical variation. Robinson [6] pointed out that factors from the resource-directing and resource-dispersing dimensions would produce interactive effects, but, unfortunately, he did not examine this further. The present study has made contributions in this respect, although more studies are needed.

## 5. Conclusions

This study investigated the effects of writing task complexity on linguistic complexity by manipulating task complexity using the number of elements and prior knowledge, as well as measuring linguistic complexity with 39 indices. The major findings are as follows: (1) increasing writing task complexity, with respect to the number of elements, significantly increased the length of production units, the number of coordinates, sentence complexity, lexical density, the number of different words, and type-token ratio but, at the same time, significantly reduced the number of particular structures, lexical sophistication, and verb variation; (2) increasing writing task complexity, with respect to prior knowledge, significantly increased the number of different words but, at the same time, significantly reduced the lexical sophistication and verb variation; (3) the number of elements and prior knowledge produced no interactive effect on syntactic complexity, lexical density, or lexical sophistication, but they significantly interacted in lexical variation. When the learner had less relevant prior knowledge of the writing topic and there were fewer elements involved in the task, the lexical variation was higher; when the learner had more relevant prior knowledge and there were fewer elements involved in the task, the lexical variation was lower.

Different from most of the existing studies on task complexity, this study did not simultaneously analyze linguistic complexity, accuracy, and fluency. Instead, it examined linguistic complexity in a more comprehensive way. Study results indicate that linguistic complexity is a multi-dimension construct, and its various sub-dimensions react differently to the increases in task complexity. Linguistic complexity would only be improved after

targeted training in the long term. Teachers should fully understand this and then design and sequence English writing tasks based on their students' English writing proficiency, teaching objectives as well as the nature of the tasks. For example, if students seldom use advanced words in their English writing, teachers could employ tasks that assume more prior knowledge, which has been proved to be helpful in improving lexical sophistication. If the teaching objective is to encourage students to use more particular structures, such as complex nominals or adjective phrases, teachers could turn to tasks with fewer elements, which would not consume so much of their students' attentional capacity, allowing them to pay more attention to language.

Understandably, this study has its own limitations. For example, it compared English proficiency among the four groups of participants before collecting data. However, considering that this study investigated writing tasks, it might be more appropriate to compare English writing proficiency among the four groups. Moreover, each group consisted of both English and non-English majors, and the study provided two separate scores of these two majors for proficiency comparison. However, it might be better to offer one score for the whole group.

Finally, it should be pointed out that this study only examined linguistic complexity comprehensively. Future studies could turn to not only linguistic accuracy and fluency but also content and organization [46–48] in greater depth, further exploring the relationships among task complexity, attentional capacity, and language output. Moreover, this study did not pay enough attention to task performers, whose emotion, motivation, and perception might moderate the effects of task complexity [49]. Future studies can make contributions in this respect.

**Supplementary Materials:** The following supporting information can be downloaded at: https://www.mdpi.com/article/10.3390/su14084791/s1.

**Author Contributions:** Conceptualization, L.W.; Data curation, L.W.; Formal analysis, L.W. and C.J.; Methodology, L.W. Writing—original draft, C.J. and L.W.; Writing—review and editing, C.J. and L.W. All authors have read and agreed to the published version of the manuscript.

**Funding:** This research received no external funding.

**Institutional Review Board Statement:** The study was conducted in accordance with the Declaration of Helsinki, and approved by the Ethics Committee of School of Foreign Languages, Nanchang University, China.

**Informed Consent Statement:** Informed consent was obtained from all subjects involved in the study.

**Data Availability Statement:** The datasets generated and/or analyzed during the current study are available from the first author on reasonable request.

**Conflicts of Interest:** The authors declare no conflict of interest.

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
