# Peer review of "Effects of Task Complexity on Linguistic Complexity for Sustainable EFL Writing Skills Development"

_sustainability, doi:10.3390/su14084791_

Round 1
Reviewer 1 Report
Comments:
This study examined the effects of task complexity (number of elements and prior knowledge) on linguistic complexity in EFL writing. The study found that task complexity features produced different effects on different dimensions of linguistic complexity. The study offers pedagogical implications and makes theoretical contribution to the field of second language writing by incorporating various linguistic complexity measures of EFL writing. However, I have a few concerns on the literature review and research design as well as procedure for statistical analysis and hope these comments would offer some help in the authors’ revision of the paper.
- Introduction
The author may add some comments on the two task complexity features and associate them with the discussion on the two theories to highlight the importance of the study.
- Literature review
The subtitle of 2.1 “studies on few elements of writing tasks” may not be appropriate. As far as I know, this feature is usually referred to as “number of elements”.
Overall, the literature review lacks a clear focus. The review of studies provides details regarding each study, but there needs to be a synthesis of previous studies and comments on the similarities and differences in research findings of these two task complexity features to establish a strong rationale for the study. Although at the end of this section, the authors comment on the reasons underlying these mixed findings, more reasons need to be provided as to why the authors selected these two features and how this study may contribute to current literature on task complexity and second language writing. In addition, recent meta-analysis may be reliable resources to find the influence of task complexity features on L2 writing and the authors may consider adding results from meta-analysis to the literature review.
- Research design
The research design needs more clarification. The participants include both English majors and non-English majors. The authors seem to have differentiated these two groups and presented ANOVA results of their proficiency scores separately. But from the description of data analysis, these two groups’ performance were pooled to find out differences in terms of two task complexity features. Therefore, the four groups (each with English and non-English majors) should be compared in the aspect of proficiency. P.6, line 236 “After confirming that the four groups of English majors and non-English majors were homogeneous”, here the authors should provide proficiency indices that reflect the whole group instead of separate scores for English and non-English majors.
How did the authors operationalize "prior knowledge" in the task? How did the authors measure “familiarity with the topic”? The article did not provide clear information on this. It is better to add a table detailing the operationalization of the task complexity features.
The description of the research design is a bit confusing. How did each group complete the tests? The tasks together with the instructions should be provided in the Appendix. Table 3. 2 Data Collection Arrangement should be reorganized to align with data analysis. The data collection included two phases, during these few months, would some other factors, such as instruction and practice, lead to the improvement in L2 writing performance reflected in syntactic and lexical complexity? How would the authors control other variables?
- Data analysis and results
On p.6, the authors stated that they included 14 indices for syntactic complexity and 25 for lexical complexity, but the results section only reported 4 syntactic complexity measures and 11 lexical complexity measures. Could the authors justify this selection and comment on the non-significant ones as well?
Regarding data analysis, the authors stated that a two-factor mixed design was adopted, with “the within-subject factor being the "few elements", and the between-subject factor being the "prior knowledge"”. Could the authors justify why they assign one factor as within-subject factor and the other as “between-subject factor” given that these two factors are similar in nature? Also, repeated measures ANOVA requires at least 3 levels for the within-subject factor, but each of these two factors only has 2 levels. In addition, other assumptions of repeated measures ANOVA need to be met, such as normality, sphericity, homogeneity of variance etc. Could the authors provide these statistics before they report results of referential statistical analysis? After the ANOVA results, the authors ought to provide post-hoc analysis to find out where the differences lie.

Author Response
- Introduction
The author may add some comments on the two task complexity features and associate them with the discussion on the two theories to highlight the importance of the study.
We have added the reasons and some comments for examining the two task complexity features in our study and highlighted the importance of the study by explaining its significance.
- Literature review
- The subtitle of 2.1 “studies on few elements of writing tasks” may not be appropriate. As far as I know, this feature is usually referred to as “number of elements”.
“Few elements” is the term used by Robinson in his Triadic Componential Framewok (2007). That’s why we used the term in my study. However, it is also true that some researchers preferred to use “number of elements”. To be in line with popular studies, I changed the expression to “number of elements”, not only here in the subtitle, but also other places in the paragraphs.
- Overall, the literature review lacks a clear focus. The review of studies provides details regarding each study, but there needs to be a synthesis of previous studies and comments on the similarities and differences in research findings of these two task complexity features to establish a strong rationale for the study. Although at the end of this section, the authors comment on the reasons underlying these mixed findings, more reasons need to be provided as to why the authors selected these two features and how this study may contribute to current literature on task complexity and second language writing. In addition, recent meta-analysis may be reliable resources to find the influence of task complexity features on L2 writing and the authors may consider adding results from meta-analysis to the literature review.
First of all, your suggestion of a synthesis of previous studies and comments on the similarities and differences in research findings of these two task complexity features is of great help for the paper, so we added a summary of the similarities and differences, especially the differences, before the comments on the reasons underlying the mixed results. Moreover, we also explained why we selected number of elements and prior knowledge, but not here, in the last paragraph of the introduction part instead, together with Johnson’s synthesis and meta-analysis of cognitive writing task complexity in 2017. However, we did have a review the related study in details in the first part, compare the results in our study to those of the previous studies and analyze the reasons that might account for the differences in the results and discussion section. Nevertheless, we could read more about recent meta-analysis for the future study, and we do appreciate the suggestion.
- Research design
- The research design needs more clarification. The participants include both English majors and non-English majors. The authors seem to have differentiated these two groups and presented ANOVA results of their proficiency scores separately. But from the description of data analysis, these two groups’ performance were pooled to find out differences in terms of two task complexity features. Therefore, the four groups (each with English and non-English majors) should be compared in the aspect of proficiency. P.6, line 236 “After confirming that the four groups of English majors and non-English majors were homogeneous”, here the authors should provide proficiency indices that reflect the whole group instead of separate scores for English and non-English majors.
It was true that some proficiency indices that reflect the whole group would be better for the English proficiency comparison among the four groups consisting both English and non-English majors. Therefore, in the revised paper, we explained why we did not organize a test for both English and non-English majors before data collecting, and listed it as one of limitations of our study.
- How did the authors operationalize "prior knowledge" in the task? How did the authors measure “familiarity with the topic”? The article did not provide clear information on this. It is better to add a table detailing the operationalization of the task complexity features.
After careful thinking of the instrument section, we found it’s true that we had not made it very clear as to how “prior knowledge” was operationalized in our study, so we have made the revision in the article. Since there had been five tables in our article, we did not add one more in case that too many tables may distract readers.
- The description of the research design is a bit confusing. How did each group complete the tests? The tasks together with the instructions should be provided in the Appendix. Table 3. 2 Data Collection Arrangement should be reorganized to align with data analysis. The data collection included two phases, during these few months, would some other factors, such as instruction and practice, lead to the improvement in L2 writing performance reflected in syntactic and lexical complexity? How would the authors control other variables?
An appendix of the four writing tasks together with the instructions was provided. In the 4 months between the two phases of data collection, there is a two-month summer vacation, from July to August. The interval will actually help control the effect of variables like instruction and practice, because a majority of participants will take a rest from their school education during the summer vacation. We had not stated it clearly, and we have made the revision in the article.
- Data analysis and results
1) On p.6, the authors stated that they included 14 indices for syntactic complexity and 25 for lexical complexity, but the results section only reported 4 syntactic complexity measures and 11 lexical complexity measures. Could the authors justify this selection and comment on the non-significant ones as well?
The measures in Table 4.1 and 4.2 were reported because they were significantly influenced by at least one independent variable according to data analysis. However, we should be careful enough since we had included one measure that had not been influenced significantly by either of the two independent variables.
It is a good suggestion to comment on the non-significant measures, but that is not the focus of this paper. We added Appendix 2 reporting the data of all measures.
2)Regarding data analysis, the authors stated that a two-factor mixed design was adopted, with “the within-subject factor being the "few elements", and the between-subject factor being the "prior knowledge"”. Could the authors justify why they assign one factor as within-subject factor and the other as “between-subject factor” given that these two factors are similar in nature? Also, repeated measures ANOVA requires at least 3 levels for the within-subject factor, but each of these two factors only has 2 levels. In addition, other assumptions of repeated measures ANOVA need to be met, such as normality, sphericity, homogeneity of variance etc. Could the authors provide these statistics before they report results of referential statistical analysis? After the ANOVA results, the authors ought to provide post-hoc analysis to find out where the differences lie.
Number of elements and prior knowledge are not similar in nature, because the former belongs to the resource-directing dimension while the latter the resource-dispersing dimension. In fact, these two factors were randomly assigned to be within-subject and between-subject factor. We should have made it clear. We have made the revision in the first paragraph of 3.3 Data collection and analysis.
We referred to books about statistical analysis and found that repeated measures ANOVA could also be applied when there are two levels for the within-subject factor, and when the within-subject factor has only two levels, there will be no sphericity reported. This is the case in our study. We found no sphericity, but the epsilon provided by Greenhous-Geisser and Hyunh-Feldt in our study for all the language measures was one, indicating that the sphericity assumption was observed. Moreover, because there are only two levels, it is not necessary to conduct the post-hoc analysis.
We also reread relative literature running repeated measures ANOVA and MANOVA, and found that many of them (e.g. Frear & Bitchener 2015; Cho 2019) did not report the exact statistics for normality, although they have checked it. Partly due to the fact that there are usually many language measures employed in the study, it will be a long list to report all the statistics. For example, in our study, there will be 78 values for normality altogether. That’s why we did not report the statistics in this respect. However, thanks to your reminder, we added the explanation that we had checked normality before reporting the results in the article.
Reviewer 3
The only thing that I can say is that I would have explained more the indices for rating the complexity (the 14 syntactic ones and the 25 covering lexical stuff). Researchers might be familiar with those indices, but it would be useful to have a deeper explanation for teachers. Maybe a new 3.3.1 section.
It is true that we did not explain all the 39 indices one by one, since it would take up a huge space if we did so. We do appreciate your suggestion in advising us to explain more about them. We

Reviewer 2 Report
Please, see the attachment.

Author Response
We have revised and please see attached document.

Reviewer 3 Report
It has a clear introduction with enough references to the main theories (limited attentional capacity hypothesis and Cognition hypothesis) that can be related with this topic.
The main idea of the research about complexity of the task is a great one, very interesting and relevant for both researchers in Applied linguistics, but also teachers. In fact, one of the strengths of this paper is that it is well and clearly written and explained. It is easy to follow the reasoning.
The authors consider possible explanations for the results of previous works with the schema theory in mind, and also if the topic of the task is familiar or not.
The sample is big enough (175 students) and the methodology is clearly explained. The data analysis is accurate and appropriate. The options considered are also well designed: task 1 with fewer elements and more related to prior knowledge, the 2 fewer elements and less prior knowledge, the 3 more elements and more prior knowledge and the fourth the hardest one: more elements and less prior knowledge.
The only thing that I can say is that I would have explained more the indices for rating the complexity (the 14 syntactic ones and the 25 covering lexical stuff). Researchers might be familiar with those indices, but it would be useful to have a deeper explanation for teachers. Maybe a new 3.3.1 section. This is because in the analysis terms like lexical sophistication are used, but it is not until page 9 where the explanation about what it is is done. Of course, as a researcher on this area (vocabulary) I know some of them and might wonder the other ones, but it would be easier for language teachers to have this more explained.
The conclusions are consistent and clearly derived from the data analysis.
The writing style is good as far as I can tell because I am not native speaker of English, but despite this, I understand everything perfectly.
Author Response
The only thing that I can say is that I would have explained more the indices for rating the complexity (the 14 syntactic ones and the 25 covering lexical stuff). Researchers might be familiar with those indices, but it would be useful to have a deeper explanation for teachers. Maybe a new 3.3.1 section.
It is true that we did not explain all the 39 indices one by one, since it would take up a huge space if we did so. We do appreciate your suggestion in advising us to explain more about them. We have made the revision in the article.

Round 2
Reviewer 2 Report
Because I rejected the paper in the first round of review, I do not feel like reveiwing it again.
Author Response
please see the attachment

This manuscript is a resubmission of an earlier submission. The following is a list of the peer review reports and author responses from that submission.